# Recent Advances on Nitrogen Use Efficiency in Rice

Sichul Lee 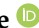

Center for Plant Aging Research, Institute for Basic Science (IBS), Daegu 42988, Korea; scironlee@gmail.com

**Abstract:** Rice (*Oryza sativa* L.) is a daily staple food crop for more than half of the global population and improving productivity is an important task to meet future demands of the expanding world population. The application of nitrogen (N) fertilization improved rice growth and productivity in the world, but excess use causes environmental and economic issues. One of the main goals of rice breeding is reducing N fertilization while maintaining productivity. Therefore, enhancing rice nitrogen use efficiency (NUE) is essential for the development of sustainable agriculture and has become urgently needed. Many studies have been conducted on the main steps in the use of N including uptake and transport, reduction and assimilation, and translocation and remobilization, and on transcription factors regulating N metabolism. Understanding of these complex processes provides a base for the development of novel strategies to improve NUE for rice productivity under varying N conditions.

**Keywords:** rice; nitrogen use efficiency; fertilization; productivity

## 1. Introduction

Rice is one of the major cereal grains to feed half of the world's population as a key nutritional source [1]. Accordingly, improving the production of rice is indispensable to satisfy the rising demands driven by population growth. The global population is expected to reach 10 billion, which will require a 70–100% increase in global crop production by 2050 [2]. The macronutrient N is an essential element for all living organisms because the synthesis of numerous vital biomolecules such as amino acids, proteins, nucleic acids, chlorophyll, and some plant hormones relies on N availability [3]. Thus, the application of N fertilizer has become a major factor responsible for the increase in crop yield over the past five decades, whereas excess usage of fertilization has caused serious environmental and economic issues [4]. The average nitrogen use efficiency (NUE) in crops is about 40–50% of the applied N, while the rest of the applied N is lost and enters the environments as N pollution [5]. The unutilized N causes water and air pollution affecting health, damage to biodiversity. The production of N fertilizer produced by the Haber–Bosch process requires a high amount of energy and contributes to greenhouse gas and then to climate change [6]. Besides, it represents increasing farmers' economic costs [7].

Improvement of NUE is one of the main objectives of breeding for rice that efficiently uptakes, assimilates, and remobilizes all available N resources [8]. Rice cultivars with high NUE are not only expected to increase grain yield but also to reduce environmental costs and facilitate low-input sustainable rice cultivation. Consequently, understanding the various processes of N metabolism is critical for increasing NUE. In this review, I will try to delineate the current knowledge on the following aspects of N metabolism: definition of NUE; N source and uptake; N assimilation and reutilization; N signaling and transcriptional regulation; and future perspectives.

## 2. What Is the Nitrogen Use Efficiency?

NUE is a complex trait affected by the interaction of multiple biochemical pathways and environmental factors. For plant physiologists, there are several definitions and equations of NUE [9] that take into account several steps in N management such as

uptake, assimilation, allocation, and remobilization especially during senescence, and can be divided into different components as N uptake efficiency (NUpE), N assimilation efficiency (NAE) and N remobilization efficiency (NRE) [6]. For agronomists, NUE has been defined as seed yield relative to the amount of N available from the soil, including N fertilizer [10]. Globally both physiologists and agronomists agree that NUE comprises two key components, N uptake efficiency (NUpE) and N utilization efficiency (NUtE), which can be monitored in the field as in the lab [6]. NUpE is the capacity to absorb, or uptake, supplied N from the soil, and NUtE is the capacity to utilize N within the plant to assimilate and remobilize to produce the harvested product such as grains. NUtE is the optimal harmonization between N assimilation efficiency (NAE) and N remobilization efficiency (NRE) [11]. NAE is the capacity to assimilate inorganic N to produce amino acids and other essential N-containing molecules, and NRE depends on the amount of N remobilized from source-to-sink tissues [11]. Each component is relevant to diverse and complex traits including root morphological features, the capacity of extracting available N from the soil, leaf senescence, and N remobilization [12]. Strategies to improve NUE in rice have mainly involved in the genetic modification of N uptake, N assimilation, and remobilization and their regulations [7]. In addition, conventional breeding programs, agronomic practice, and a combination of them are also critical to increase NUE [12].

## 3. Nitrogen Source and Uptake

Plants have been developed several N uptake systems to cope with the fluctuating and challenging environment because plants are non-mobile. N is first taking up from the soil by plant roots as ammonium ($NH_4^+$) and nitrate ($NO_3^-$) [13]. Ammonium is the main form in flooded paddy fields due to the anaerobic soil condition, while nitrate is abundant in naturally aerobic upland owing to intensive nitrification from applied organic and fertilized N [13]. Urea is one of the commonly used fertilizers and it could be degraded into ammonium and carbon dioxide by urea-degrading enzymes that are secreted by microorganisms in soil [14]. However, few reports have shown that urea can be imported from the environment into root cells [14,15]. The spray of urease inhibitors in the field is a technical solution to improve urea use by plants [16]. Amino acid also can be used by plants in soils and amino-N is utilized for a variety of metabolic processes [17]. The first step of N acquisition by plant roots is active influx by the transmembrane transporters present in root epidermal and cortical cells. Current approaches for N transportation are summarized in Table 1 and Figure 1.

### 3.1. Nitrate Transporters in Rice

Transmembrane transporters are necessary for nitrate absorption from the soil and inter-and intracellular transportation and translocation inside the plants. About 40% of the total N taken up by rice is absorbed as nitrate because of the nitrification in the rhizosphere [18]. Four protein families including nitrate transporter 1 (NRT1)/peptide transporter (PTR) family (subsequently known as unified nomenclature NPF), nitrate transporter 2 (NRT2) family, chloride channel (CLC) family, and slow anion channel-associated homologs (SLAC/SLAH) family are involved in nitrate transport [19,20]. In rice, there are 94 NPF [21,22], 5 NRT2 [23], 5 CLC [24], and 9 SLAC1/SLAH [25] members.

Only small numbers of NPF (NRT1/PTR family) have been characterized to date (Table 1) [26]. OsNPF4.1 (SP1) determines the development of the rice panicle size, while transporter activities using nitrate, dipeptides, His and carboxylate could not be detected in either *Xenopus* oocytes or yeast system [27]. OsNPF8.9 (OsNRT1.1) is a low-affinity nitrate transporter as shown by a *Xenopus* oocyte assay system [28]. The expression of two splicing variants of *OsNRT1.1* was downregulated by N deficiency in roots, and overexpression of *OsNRT1.1a* showed increased N accumulation only under high N supply. However, *OsNRT1.1b* overexpression lines increased total N concentration and improved rice growth under low and high N conditions [29]. Vacuolar membrane-localized OsNPF7.3 (OsPTR6) transporter can transport dipeptide Gly-His and the tripeptide Gly-Gly-Leu [30]. Its overex-

pression promotes rice growth, by increasing N uptake and glutamine synthetase activity, but decreases NUE under conditions of high ammonium supply [31]. However, in the paddy field, *OsNPF7.3* overexpressing plants showed increased tiller and panicles numbers per plant, better fertility, and higher grain N content, thus indicating improved NUE. RNAi lines displayed the opposite phenotypes [32]. Expression of *OsNPF8.20* (*OsPTR9*) is regulated by exogenous N and photoperiod. Heterologous expression systems could not reveal any transporting activities for di/tripeptides, nitrate, carboxylates, or ammonium [33]. Nevertheless, *OsNPF8.20* overexpressing lines resulted in enhanced ammonium uptake and improved grain yield in both normal and no N fertilizer. Consistently, knockout or RNAi lines displayed the opposite effects [33]. *OsNPF2.2* (*OsPTR2*) is a plasma membrane-localized low-affinity and pH-dependent nitrate transporter [34]. The *osnpf2.2* mutants accumulate higher levels of nitrate in their roots than in shoots, and display retarded plant growth and development, with abnormal vasculature. OsNPF2.4 functions in the low-affinity acquisition and long-distance transport of nitrate [35]. Its disruption impairs potassium coupled nitrate upward transport from the root to the shoot.

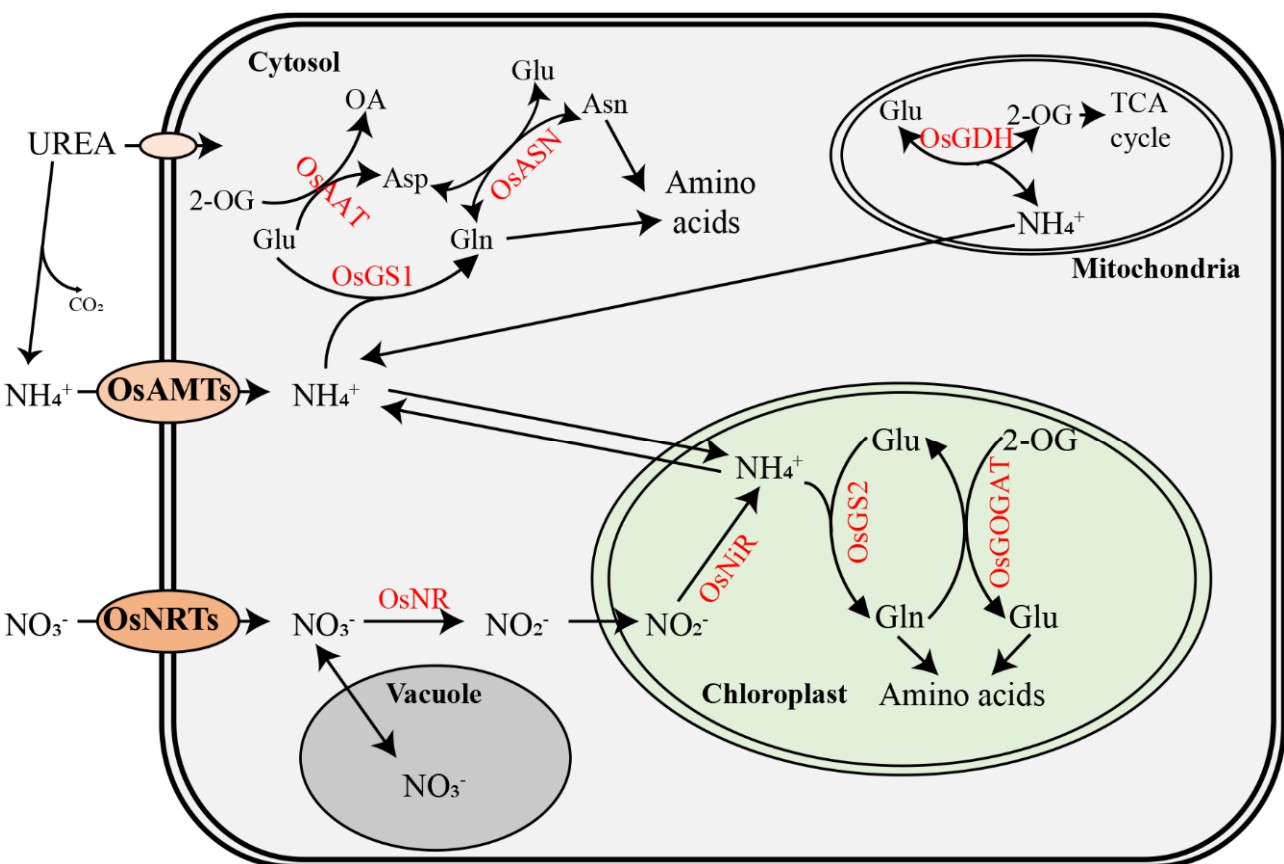

**Figure 1.** Schematic representation of key processes and enzymes involved in N metabolisms in rice. Ammonium and nitrate ions were taken up by ammonium transporters (OsAMTs) and nitrate transporters (OsNRTs) across the cell membrane. Urea is directly absorbed by transporters or degraded into ammonium ions. Nitrate is reduced into nitrite by nitrate reductase (OsNR) in the cytosol and then reduced into ammonium by nitrite reductase (OsNiR) in the plastids. Ammonium is assimilated into N-containing compounds via the glutamine synthetase (GS)/glutamine-2-oxoglutarate aminotransferase (GOGAT) cycle, and then further assimilated for N metabolism through asparagine synthetase (ASN), glutamate dehydrogenase (GDH), and aspartate aminotransferase (AAT). Glu, glutamate; Gln, glutamine; Asn, asparagine; Asp, aspartate; 2-OG, 2-oxoglutarate; OA, oxaloacetate. All these steps are regulated by signaling molecules and transcription factors.

OsNPF6.5 (NRT1.1B) diverged between *indica* and *japonica* during rice domestication, and variations in NRT1.1B-*indica* contributed to enhanced nitrate uptake, to better root-to-

shoot transport, and upregulated expression of nitrate-responsive genes [36]. *Japonica* near-isogenic (NIL) and transgenic lines carrying the NRT1.1B-*indica* allele exhibited significantly improved grain yield and NUE. In addition, N uptake assays using *Xenopus* oocytes showed that the NRT1.1B-*indica*-injected oocytes exhibited higher nitrate uptake activity than the NRT1.1B-*japonica*-injected oocytes. This demonstrated that NRT1.1B-*indica* has higher activity than NRT1.1B-*japonica* [36]. OsNPF7.2 is a tonoplast localized low-affinity nitrate transporter and its knock-down plants display retarded growth phenotype under high nitrate supply [37]. OsNPF8.1 (OsPTR7) displays transport activity for dimethyarsinate (DMA) in *Xenopus* oocytes and is involved in the long-distance translocation and the accumulation of DMA in rice grain [38]. OsNPF7.7 has two splicing variants which showed different expression patterns and different cellular localizations. The long variant *OsNPF7.7-1* is highly expressed in panicles and localized at the plasma membrane; the short-variant *OsNPF7.7-2* is highly expressed in buds and leaves and localized at the vacuole membrane [26]. Overexpression of *OsNPF7.7* promotes N influx in root and improves grain yield NUtE. OsNRT1.1A (OsNPF6.3) displayed ammonium-inducible expression and localized to the tonoplast [20]. The *osnrt1.1a* mutants display late flowering phenotype and reduced N utilization. On the contrary, the *OsNRT1.1A* overexpressing lines display better N utilization and grain yield than the wild type, with shortened maturation time [20]. OsNPF6.1 was isolated by GWAS on rice landraces with extreme N-related phenotypes [39]. It showed nitrate transport activity in *Xenopus* oocyte heterologous expression system. Its natural allele OsNPF6.1$^{HapB}$ derived from variations in wild rice enhances nitrate uptake and confers high NUE by increasing yield under low N supply.

NRT2 family represents a high-affinity transport system (HATS), and NAR2, a putative partner of NRT2 proteins, is required for the functionality of the HATS [23,40]. OsNAR2.1 interaction with OsNRT2.1, OsNRT2.2, and OsNRT2.3a is needed to address transporters to the plasma membrane and to maintain the stability of NRT2 proteins and thus uptake activities. The OsNRT2.3b and OsNRT2.4 transporters do not require OsNAR2 [41,42]. OsNRT2.3a is localized to the plasma membrane and is mainly expressed in the xylem parenchyma cells of the stele of nitrate-supplied roots [23]. Its RNAi lines display impaired nitrate loading in xylem saps and decreased plant growth at low nitrate supply. OsNRT2.3b is located on the plasma membrane and expressed in the phloem [43]. Overexpression of *OsNRT2.3b* enhances the pH-buffering capacity of nitrate uptake and improves grain yield and NUE. Plants overexpressing *OsNRT2.1* grow faster under nitrate supply, but no increase in nitrate uptake could be observed [44]. Increased expression of *OsNRT2.1*, driven by the *OsNAR2.1* promoter, improved yield and NUE [13]. In another study, overexpression of *OsNRT2.1* also improved grain yield, and also grain manganese content under water-saving cultivation [45]. It was also shown to enhance the nitrate-dependent root elongation [46].

Plant CLCs transporters are critical for uptake and transport of Cl$^-$ and nitrate under salt stress [47,48]. The rice genome contains five *CLC* genes, but their functional roles for nitrate transport have not been reported so far [24]. Instead, the tonoplast-localized OsCLC1 and OsCLC2 proteins function in compartmentalizing Cl$^-$ into the vacuole to avoid the toxicity of Cl$^-$ under salt stress conditions, and overexpression of *OsCLC1* promotes drought tolerance, leading to better grain yield [24,49].

### 3.2. Ammonium Transporters Rice

There are at least 12 putative ammonium transporter families which could be divided into five groups in rice [50]. Current knowledge of OsAMT transporters is summarized in Table 1. OsAMT1;1 is a proton-independent ammonium transporter as shown by the functional complementation of yeast mutant in combination with the electrophysiological assay in *Xenopus* oocytes [51]. OsAMT1;1 is localized in the plasma membrane and constitutively expressed in shoots and roots [52]. Disruption of *OsAMT1;1* reduces ammonium uptake and seedling growth as it decreases N transport from roots to shoots [52]. Its overexpression in transgenic plants enhanced N uptake and plants displayed greater N assimilation, improved plant growth, and higher grain yield, especially under suboptimal

$NH_4^+$ [53]. By contrast, the study of Hoque et al. showed that the enhanced expression of *OsAMT1;1* has a poor effect on plant growth during seedling and early vegetative stage [54]. This may be due to unbalance between the enhanced ammonium uptake and the unchanged ammonium assimilation. Expression of *OsAMT1;2* is root-specific and ammonium inducible [55]. Increasing its expression using the *35S* enhancer resulted in better ammonium uptake and increased tolerance to N limitation at the seedling stage. However, impaired growth and lower grain yield, with higher grain N and protein content, were observed when plants were grown in the paddy field [56]. *OsAMT1;3* is root-specific and induced by N starvation [55]. *OsAMT1;3* overexpressing transgenic lines display poor N uptake ability. Their growth is decreased, and plants display a higher C/N ratio [57].

Regarding the OsAMT2 transporters, very few are known. OsAMT2;1 is constitutively expressed in both roots and shoots, and it functionally complements the ammonium-uptake-defective yeast mutants [55]. *OsAMT2;2* is also expressed in both root and shoot [55]. *OsAMT3;1* shows very weak expression in root and shoot [58]. Still, nothing is known about the exact function of other OsAMTs in rice.

### 3.3. Urea Transporters in Rice

Urea can be used as a readily available N source for plant growth and has become the frequently used form of N fertilizer in agriculture due to its low production costs and high N content [14]. Rice urea transporter, OsDUR3 is membrane-localized and mediates a high-affinity urea import in *Xenopus* oocytes [59]. Its expression is upregulated under N deficiency and in urea-supplied rice roots after N starvation [59]. Knockout lines by *Tos17* insertion showed yield reduction in both hydroponic culture and a paddy field with decreased grain filling, indicating that OsDUR3 is involved in N distribution to panicles during heading [15].

### 3.4. Amino Acid Permeases

Different families of transporters for amino acid uptake and translocation with different substrate specificities and affinity include amino acid permeases (AAPs), lysine-histidine-like transporters (LHTs), proline transporters (ProTs), γ-aminobutyric acid (GABA) transporters (GATs), ANT1-like aromatic, and neutral amino acid transporters and cationic amino acid transporters (CATs) [60]. AAPs are considered to be moderate-affinity systems with broad substrate specificity [61]. In rice, nineteen AAP genes were identified [60]. The quantitative trait locus *qPC1*, which encodes the amino acid transporter OsAAP6, functions as a positive regulator of grain protein content (GPC) [62]. The higher transcript levels of *OsAAP6*, due to a variation in the 5′-untranslated region of *indica* cultivars, enhances uptake of Thr, Ser, Gly, Ala, Pro, and acidic amino acids, and leads to higher GPC. Overexpression of *OsAAP3* results in higher concentrations of Lys, Arg, His, Asp, Ala, Gln, Gly, Thr, and Tyr [63]. It also decreases bud outgrowth, tillering, and grain yield, while RNAi of *OsAAP3* promotes bud outgrowth, increased tiller, and effective panicle numbers per plant, leading to enhanced grain yield and NUE. The lower expression of *OsAAP5* in *indica* varieties compared to *japonica* varieties was associated with less tillers in *japonica* [17]. The *OsAAP5* RNAi lines in *japonica* produced more tillers and get better grain yield by decreasing the contents of basic amino acids (Lys and Arg) and neutral amino acids (Val and Ala). In contrast, the overexpression of *OsAAP5* showed the opposite effects.

*OsAAP1* is highly expressed in rice axillary buds, leaves, and young panicles [64]. OsAAP1 protein is localized to both the plasma membrane and the nuclear membrane. Overexpression of *OsAAP1* led to higher concentrations of neutral and acidic amino acids and increased grain yield as a result of better grain filling and tillering. The RNAi down-regulated and CRISPR knockout lines showed the opposite phenotype. The higher expression of *OsAAP4* in *indica* varieties relative to *japonica* varieties was caused by variations in the promoter sequence [65]. This resulted in more tillers and higher grain yield in the *indica* rice. Subsequently, the overexpression of two different splicing variants of *OsAAP4* in *japonica* increased the numbers of the tiller and the grain yield, enhancing the neutral

amino acid concentrations of Val, Pro, Thr, and Leu. The RNAi and mutant lines showed the opposite trends.

Based on gene-based genetic association analysis of aspartate uptake between *japonica* and *indica* rice subspecies, *Lysine-Histidine-type Transporter 1 (OsLHT1)* was identified as a candidate gene [66]. *OsLHT1* is expressed throughout the root and in the leaf, and *oslht1* mutants showed the reduced amino acids root uptake and allocation from root to shoot, thereby growth inhibition and low yield [66–68]. Plasma membrane-localized OsProT1 and OsProT3 function to specifically mediate transport of Pro and GABA in yeast and showed differential expression patterns [69].

## 4. N Assimilation and Reutilization

Assimilation of ammonium and utilization is a complex and tightly regulated process for rice biomass production and grain yield [70]. After nitrate is absorbed into roots, it is first reduced to nitrite by nitrate reductase (NR) in the cytoplasm, and then nitrite is further reduced to ammonium by nitrite reductase (NiR) in the plastids [71]. *OsNR2* encoding a NAD(P)H-dependent nitrate reductase (NR), was isolated as the master gene causing differences in nitrate assimilation and NUE between the *indica* and *japonica* rice [71]. Variations in *indica* and *japonica OsNR2* alleles result in structurally distinct OsNR2 proteins, with higher NR activity in *indica*, conferring increased effective tiller number, grain yield, and NUE compared to *japonica* rice [71]. The ammonium from nitrate reduction and ammonium uptake by OsAMTs is assimilated into amino acid via the glutamine synthetase (GS)/glutamine-2-oxoglutarate aminotransferase (GOGAT) cycle (Figure 1 and Table 2) [72].

### 4.1. Glutamine Synthetase

GS catalyzes an ATP-dependent conversion of glutamate (Glu) to glutamine (Gln) using ammonium and plays an essential role in the N metabolism. In rice, there are several homologous genes for the cytosolic GS (*OsGS1;1*, *OsGS1;2*, and *OsGS1;3*) and one chloroplastic gene (*OsGS2*) [73]. *OsGS1;1* is expressed in all organs and especially highly expressed in the leaf blade. Its disruption results in severe retardation of growth rate and lower grain productivity [73]. OsGS1;1 plays a critical role in maintaining the metabolic balance of rice plants grown under ammonium as an N source [73,74]. In addition, N assimilation by OsGS1;1 affects plastid development in rice roots [75]. The overexpression of *OsGS1;1* or *OsGS1;2* did not increase the grain yield or total amino acids in seeds [76].

*OsGS1;2* is mainly expressed in the surface cells of roots that are responsible for the primary assimilation of ammonium [77]. Metabolic disorders caused by the lack of *GS1;2* lead to a severe reduction of the number of active tiller and grain yield [77–79]. *OsGS1;3* was exclusively expressed in the spikelet, but its functional role for N assimilation is still unknown [73]. OsGS2 isoform is abundant in the leaf and OsGS2 is primarily responsible for re-assimilation of ammonium produced from photorespiration in chloroplasts and assimilation in plastids of ammonium deriving from nitrate reduction [73,80]. The overexpression of *OsGS2* enhanced photoprotection and thereby enhanced tolerance to abiotic stresses [81].

### 4.2. Glutamate Synthetase

GOGAT yields two molecules of Glu by transferring the amine group of the amide side chain of Gln to 2-oxoglutarate (2-OG) [82]. Subsequently, one Glu molecule serves as a substrate for GS, while the other is used for transport, storage, or further amino acid metabolism. There are two types of GOGAT using either reduced ferredoxin (Fd-GOGAT) or NADH (NADH-GOGAT) as an electron donor. In rice, there are two NADH-type GOGATs and one Fd-GOGAT [83]. *OsNADH-OsGOAT1* is mainly expressed in the roots after the supply of ammonium [84], and *OsNADH-OsGOAT1* knockout mutant displays decreased grain yield in paddy field due to the reduced number of active tillers [83]. According to similarities in their expression patterns and the phenotypes of their knock-

out mutants, it was suggested that OsNADH-OsGOAT1 and OsGS1;2 both contribute to the primary assimilation of ammonium in rice roots [84]. Enhanced expression of *OsNADH-OsGOAT1* was shown to increase N uptake and N remobilization efficiency, but the *OsNADH-OsGOAT1* overexpressing plants displayed poor growth benefits and reduced grain yield when grown in paddy field, which revealed some unbalanced use of N [56]. Nevertheless, the concomitant overexpression of both *OsNADH-OsGOAT1* and *OsAMT1;2* conferred enhanced NUE and better grain yield under N limiting growth conditions, which can provide a technical solution for plant performances under nitrate limiting conditions [56].

*OsNADH-GOGAT2* is expressed mainly in fully expanded leaf blade and sheath. Defects in *OsNADH-GOGAT2* caused a remarkable reduction in spikelet number/panicle and grain yield and whole plant biomass [70]. OsNADH-GOGAT2 along with OsGS1;1 could be important in the remobilization of N during the senescence stage [84]. *OsFd-GOGAT* is mainly expressed in the chloroplasts of green tissues as well as the mature leaf blade and sheath and plays a major role in photorespiratory ammonium assimilation [85]. Mutation of *OsFd-GOGAT* led to premature leaf senescence and disturbed C/N balance [85,86].

### 4.3. Asparagine Synthetase, Glutamate Dehydrogenase, and Aspartate Aminotransferase

In addition to GS/GOGAT, several enzymes including asparagine synthetase (ASN), glutamate dehydrogenase (GDH), and aspartate aminotransferase (AAT) were characterized to play important roles in N metabolism (Figure 1 and Table 1) [6,87,88].

Asparagine (Asn) and Gln are the major N forms in phloem sap and play a critical role in dynamic N recycling [89,90]. Asparagine synthetase (ASN) can produce asparagine by transferring the amide group from glutamine to aspartate. In *Arabidopsis*, labeling suggested that asparagine could be synthesized directly using ammonium as an N donor [91]. In rice, there are two *ASN* genes, *OsASN1* and *OsASN2* [92]. *OsASN1* is mainly expressed in roots in an ammonium-dependent manner, whereas *OsASN2* expression is decreased under ammonium supply [92]. Disruption of *OsASN1* leads to decreased N uptake and increased sensitivity to N limitation at the seedling stage. It displays reduced grain protein content and productivity, whereas the opposite tendencies are observed in overexpressing lines [93]. The mitochondrial NADH-dependent GDH catalyzes the reversible conversion of Glu to 2-OG and ammonia [94]. Three *OsNADH-GDH* genes are differentially expressed in various organs depending on N availability [95], but their roles remain unknown. AAT is involved in N and C metabolisms and catalyzes the reversible transamination of aspartate (Asp) to 2-OG, yielding Glu and oxaloacetate (OA) [88]. The overexpression of *OsAAT1* and *OsAAT2* resulted in altered N metabolism and increased amino acid content in seeds [88].

**Table 1.** Transporters and enzymes associated with N metabolic steps.

| Gene Name | Locus Number | Phenotype Observed | References |
|---|---|---|---|
| *OsNPF4.1/SP1* | LOC_Os11g12740 | Ko [1]: Defective in rice panicle elongation and the short-panicle phenotype | [27] |
| *OsNPF8.9/OsNRT1.1* | LOC_Os03g13274 | OX [2]: Increased biomass under various N supplies | [28,29] |
| *OsNPF7.3/OsPTR6* | LOC_Os04g50950 | OX: Increased growth by N accumulation but decreased NUE [5] under high $NH_4^+$ supply | [30,31] |
| | | OX: Enhanced NUE in paddy field | [32] |
| | | RNAi [3]: Decreased amino acids accumulation and plant growth | |
| *OsNPF8.20/OsPTR9* | LOC_Os06g49250 | OX: Enhanced N uptake, promotion of lateral root formation, and increased grain yield | [33] |
| | | Ko & RNAi: The opposite effects of OX | |

**Table 1.** *Cont.*

| Gene Name | Locus Number | Phenotype Observed | References |
|---|---|---|---|
| *OsNPF2.2/OsPTR2* | LOC_Os12g44100 | Ko: Reduction in root-to-shoot nitrate transport and abnormal vasculature development | [34] |
| *OsNPF2.4* | LOC_Os03g48180 | OX: Enhanced nitrate acquisition and upward transfer to shoot | [35] |
| | | Ko: The opposite effects of OX | |
| *OsNPF6.5/NRT1.1B* | LOC_Os10g40600 | NIL [4], OX: Increased grain yield and NUE | [36] |
| *OsNPF7.2* | LOC_Os02g47090 | Ko & RNAi: Retarded growth under high nitrate supply | [37] |
| *OsNPF8.1/OsPTR7* | LOC_Os01g04950 | Ko: Less accumulation of dimethyarsinate in rice grain | [38] |
| *OsNPF7.7* | LOC_Os10g42870 | OX: Improved N influx in root, grain yield, and NUtE | [26] |
| *OsNRT1.1A/OsNPF6.3* | LOC_Os08g05910 | OX: Early maturation and improved N utilization and grain yield | [20] |
| | | Ko: Reduced N utilization and late flowering | |
| *OsNPF6.1* | LOC_Os01g01360 | NIL: Enhancement of N uptake, NUE, and grain yield under low N supply | [39] |
| *OsNRT2.3a* | LOC_Os01g50820 | RNAi: Defect in long-distance nitrate transport from root to shoot | [23] |
| *OsNRT2.3b* | LOC_Os01g50820 | OX: Improved growth and NUE | [43] |
| *OsNRT2.1* | LOC_Os02g02170 | OX: Fast growth under nitrate supply, but no increase in nitrate uptake | [44] |
| | | OX by *OsNAR2.1* promoter: Increased grain yield and NUE | |
| | | OX: Increased grain yield and grain Mn under alternating wet and dry condition | [45] |
| | | OX: Enhanced the nitrate-dependent root elongation | [46] |
| *OsCLC1* | LOC_Os01g65500 | OX: Enhanced salt tolerance and grain yield | [24,49] |
| | | Ko: The opposite effects of OX | |
| *OsAMT1;1* | LOC_Os04g43070 | Ko: Decreased N uptake and the growth of roots and shoots | [52] |
| | | OX: Improved NUE and grain yield | [53] |
| | | OX: Decreased biomass at early stages of growth | [54] |
| *OsAMT1;2* | LOC_Os02g40730 | Ac: Increased tolerance to N limitation at the seedling stage, but decreased grain yield | [56] |
| *OsAMT1;3* | LOC_Os02g40710 | OX: Decreased growth with poor N uptake ability with a higher leaf C/N ratio | [57] |
| *OsDUR3* | LOC_Os10g42960 | Ko: Decreased grain filling and grain yield | [15] |
| *OsAAP6* | LOC_Os01g65670 | NIL, OX: Increased grain protein content | [62] |
| *OsAAP3* | LOC_Os06g36180 | OX: Decrease in tiller number and grain yield | [63] |
| | | Ko: Increased grain yield due to increased bud outgrowth and numbers of tillers | |
| *OsAAP5* | LOC_Os01g65660 | OX: Less tiller number and grain yield | [17] |
| | | RNAi: Increase in tiller number and grain yield | |

**Table 1.** *Cont.*

| Gene Name | Locus Number | Phenotype Observed | References |
|---|---|---|---|
| *OsAAP1* | LOC_Os07g04180 | OX: Increased, growth, tillering, and grain yield, higher concentrations of amino acids | [64] |
| | | Ko: Inhibition of axillary bud outgrowth and reduced tiller number<br>Ko & RNAi: The opposite effects of OX | |
| *OsAAP4* | LOC_Os12g09300 | OX: Increased rice tillering and grain yield | [65] |
| *OsLHT1* | LOC_ Os08g03350 | Ko: Reduced amino acids uptake and allocation, growth inhibition, and low yield | [66–68] |
| *OsNR2* | LOC_Os02g53130 | NIL: Increased effective tiller number, grain yield, and NUE | [71] |
| *OsGS1;1* | LOC_Os02g50240 | Ko: Reduction in growth rate and grain yield | [73] |
| | | Ko: Overaccumulation of free ammonium in the leaf sheath and roots | [74] |
| | | Ko: Abnormal sugar and organic N accumulation | [75] |
| | | OX: Decreases in grain yield and total amino acids in seeds | [76] |
| *OsGS1;2* | LOC_Os03g12290 | OX: Reduction in grain yield and total amino acids in seeds | [76] |
| | | Ko: Reduction in active tiller number and grain yield | [77–79] |
| *OsGS2* | LOC_Os04g56400 | OX: Enhancement of photorespiration and tolerance to salt stress | [81] |
| *OsNADH-GOGAT1* | LOC_Os01g48960 | Ko: Inhibition of the root elongation by $NH_4^+$, reduction in tiller number and grain yield | [83] |
| | | Ac: Increased N uptake and remobilization, but a reduction in grain productivity | [56] |
| *OsNADH-GOGAT2* | LOC_Os05g48200 | Ko: Reduction in Productivity Via Decrease of Spikelet Number | [70] |
| *OsFd-GOGAT* | LOC_Os07g46460 | Ko: Premature leaf senescence and reduced grain yield with higher GPC | [85] |
| | | Ko: Accumulation of an excessive amount of amino acids with disturbed C/N balance | [86] |
| *OsASN1* | LOC_Os03g18130 | Ko: Reduction in free asparagine content in roots and xylem sap | [92] |
| | | Ko: Decreased N uptake and grain yield | [93] |
| | | OX: Enhanced tolerance and grain yield under N limitation | |
| *OsAAT1* | LOC_Os02g55420 | OX: Increased amino acid content in seeds | [88] |
| *OsAAT2* | LOC_Os01g55540 | | |

[1] Knockout mutants by T-DNA or *Tos17* insertion. [2] Overexpressing transgenic lines driven by maize *Ubiquitin* or *CAMV 35S* promoter. [3] Knockdown lines by RNA interference. [4] *Japonica* cultivars harboring *indica* allele. [5] Nitrogen Use efficiency.

## 5. N Signaling and Transcriptional Regulation

In addition to transporters and enzymes, a significant number of regulatory genes involved in N metabolism have been recently reported (listed in Table 2). The maize Dof1 (ZmDof1) is a member of the Dof transcription factors specific to plants and a key regulator for the coordinated gene expression associated with the organic acid metabolism [96]. Transgenic rice overexpressing *ZmDof1* displayed induction of genes related to organic acid metabolism and increase of N assimilation resulting in better growth under low-N

conditions [97]. The overexpression of *OsRDD1/OsDof2* induced the expression of *OsGS1;1* and the increased uptake of ammonium and nitrate, and increased grain productivity [98]. It has been shown that *OsDof18* modulates ammonium uptake by inducing ammonium transporters in rice roots and mutations in *OsDof18* caused impaired growth when ammonium was the sole N source [99]. OsNAC42 can activate an elite haplotype of nitrate transporter OsNPF6.1[HapB] that enhances N absorption and confers high NUE by increasing yield under low N supply [39]. The *osnac42* mutant exhibits a lower nitrate influx rate and lower N concentration.

OsMYB305 encodes a transcriptional activator; its expression is enhanced in N deficient roots [100]. The overexpression of *OsMYB305* enhances N uptake and assimilation, and improved growth under low N, increased tiller number, shoot dry weight, and total N concentration [100]. GROWTH-REGULATING FACTOR 4 (OsGRF4) is a transcriptional regulator of multiple N metabolism genes [101]. Its activity is balanced by an antagonistic regulatory relationship with the DELLA growth repressor. Modulation of OsGRF4-DELLA balance towards increased *OsGRF4* abundance enhances NUE and grain yield. *OsMYB61* is induced by low N supply [102]. This transcription factor is directly regulated by OsGRF4, which is an integrative regulator of several N metabolism genes, as well as a coordinator of C metabolism. The natural variations at *OsMYB61* between *indica* and *japonica* subspecies mediate divergence in NUE and biomass, contributing to high NUE in *indica* [102]. Subsequently, the introduction of the *indica* allele into the elite *japonica* cultivars significantly increases the grain yield and NUE, as well as biomass production.

The MADS-box transcription factor, OsMADS25, OsMADS27, and OsMADS57 have been shown to regulate root growth and to be involved in N signaling [103–105]. *OsMADS25* and *OsMADS27* are significantly induced by nitrate treatment, and transgenic lines overexpressing them showed enhanced expression of nitrate transporter genes as well as nitrate accumulation. As a result, their phenotype showed improved root growth in the presence of nitrate [103,104]. *OsMADS57* is mainly expressed in xylem parenchyma cells of the root stele and is induced by nitrate [105]. The *OsMADS57* overexpressing lines displayed higher nitrate transporter expression levels and loading of nitrate in xylem was higher than in wild type. The *osmads57* mutants showed opposite phenotypes [105]. In addition, yeast one-hybrid and transient expression assays demonstrated that OsMADS57 binds to the CArG (CATTTTATAG) motif within the *OsNRT2.3a* promoter. This indicates that OsMADS57 plays a role in regulating nitrate translocation from root to shoot via OsNRT2.3a.

The NIN-LIKE PROTEIN (NLP) family proteins function as transcription factors in the response of several genes to nitrate-resupply by binding nitrate-responsive cis-element (NRE) [106]. *OsNLP1* is rapidly induced by N starvation in rice roots and its expression is suppressed by N resupply [107]. OsNLP1 regulates N utilization by binding the promoter of multiple N uptake and N assimilation genes to activate their expression. As a result, overexpression of *OsNLP1* improves plant growth, grain yield, and NUE under different N conditions. At the reverse, *osnlp1* mutants display lower grain yield and lower NUE under N-limiting conditions. *OsNLP4* expression is induced by N deficiency and its overexpression promoted higher grain yield and NUE under moderate N level [108]. OsNLP4 regulates and orchestrates the expression of a majority of known N uptake, assimilation, and signaling genes, by directly binding to the nitrate-responsive cis-element located in their promoters. In contrast, *osnlp4* mutants display a dramatic reduction of yield and NUE. *OsNLP4* was also identified as a key gene associated with NUE through GWAS [109]. OsNLP4 could transactivate *OsNiR* encoding nitrite reductase and enhance OsNLP4-OsNiR cascade that results in increased NUE and higher tiller number [109]. The nuclear localization of OsNLP3 leads to the activation of nitrate-responsive genes under nitrate supply, and its disruption results in severe defects in the induction of nitrate-responsive genes and N uptake ability [110].

**Table 2.** Summary of regulatory genes involved in N metabolism.

| Gene Name | Locus Number | Phenotype Observed | References |
|---|---|---|---|
| *ZmDOf1* | | OX: Enhanced N assimilation and growth under low-N conditions | [97] |
| *OsRDD1/OsDof2* | LOC_Os01g15900 | OX: Increased N transport and grain productivity | [98] |
| *OsDOF18* | LOC_Os04g58190 | Ko: Growth retardation under $NH_4^+$ supply | [99] |
| *NAC42* | LOC_Os09g32040 | Ko: Lower nitrate influx rate, and retarded growth, and reduced grain yield | [39] |
| *OsMYB305* | LOC_Os01g45090 | OX: Enhanced N uptake under low-N condition | [100] |
| *OsGRF4* | LOC_Os02g47280 | NIL: Enhanced ammonium uptake rates, N assimilation, and growth. | [101] |
| *OsMYB61* | LOC_Os01g18240 | NIL: Improved NUE and grain yield at reduced N supply | [102] |
| *OsMADS25* | LOC_Os04g23910 | OX: Enhanced root and shoot growth in a nitrate-dependent manner | [103] |
| | | Ko: Reduced shoot and root growth in the presence of nitrate | |
| *OsMADS27* | LOC_Os02g36924 | OX: Promotion of root and shoot growth under nitrate supply | [104] |
| | | Ko: Reduced shoot and root growth under nitrate supply | |
| *OsMADS57* | LOC_Os02g49840 | OX: Enhanced expression of nitrate transporters and higher levels of xylem loading of nitrate | [105] |
| | | Ko: Decreased N uptake | |
| *OsNLP1* | LOC_Os03g03900 | OX: Improved plant growth, grain yield, and NUE under different N conditions | [107] |
| | | Ko: Reduced grain yield and NUE under N-limiting conditions | |
| *OsNLP4* | LOC_Os09g37710 | OX: Increased grain yield by 30% and NUE by 47% under moderate N level | [108] |
| | | Ko: Reduced grain yield and NUE | |
| | | OX: Enhanced N assimilation efficiency and tiller number and grain yield | [109] |
| *OsNLP3* | LOC_Os01g13540 | Ko: Severe defect in the induction of nitrate-responsive genes and N uptake ability | [110] |
| *miR396ef* | | Ko [1]: Improved grain yield and NUE | [111] |

[1] Genome editing by CRISPR-Cas9 system.

Several miRNAs controlling rice yield have been identified [111]. For example, the miR396 affects panicle branching and grain yield by targeting the growth regulating factors (GRFs), which execute their function via GRF-interacting factors (GIFs) by forming a regulatory module of miR396-GRF-GIF [112]. The *osmir396ef* mutants show enhanced panicle branching and increased N assimilation and utilization [111].

## 6. Future Perspectives

N is an essential element that is required throughout the rice life cycle. Recent improvements in rice productivity were mainly due to N fertilization. Unfortunately, excessive use of N fertilizers results in serious concerns for the environment and global economy. Enhancing NUE is challenging. In recent years, a remarkable increasing number of transporters and enzymes involved in N uptake and utilization have been characterized

in rice. Signaling and transcriptional regulation networks have been dissected. As plant NUE is a complicated trait subjected to genetic factors as well as environmental clues, a better understanding of the interactions between NUE key components and agricultural management practices is required to achieve the N efficient rice. Although a substantial identification of targets for NUE improvements and several technical strategies to improve NUE in rice have been conducted during the last years, problems are still emerging, including the complex trait of N metabolism, inconsistent outcomes from the transgenic approaches, and shortfall of test in the paddy field. To solve these problems properly, continuous efforts will be required by scientists by combining molecular approaches and agricultural practices in an eco-friendly manner.

Because two representative rice subspecies, *indica*, and *japonica*, display divergence of NUE between two subspecies [71], the investigation of the natural variations in rice germplasm is an alternative way to biotechnology for NUE improvement. Interestingly, there are increasing numbers of recent reports indicating the differential nitrate uptake and assimilation capacities among rice varieties [113]. Such variability is highly encouraging the search of favorable alleles.

While most of the studies have been focused on the exploration of the mechanisms and signaling pathways involved in N metabolisms in rice, the relationship between N metabolism and phytohormone signaling in rice is largely unknown. As phytohormones are involved in almost all the biological processes in the plant, this field deserves attention.

It is well known that root-associated microbes play key roles in the acquisition and cycling of nutrients including N [114]. N fertilization can affect the soil properties and activities of the rhizosphere microbial community, which have in turn significant influences rice production [115]. In addition, information on arbuscular mycorrhizal fungi (AMF) can be considered as a crucial tool for sustainable N fertilization, which play important role in the maintenance of soil fertility, the reduction of chemical fertilizers as well as crop production [116]. Research on this area is attracting attention as a rising research hotspot, but a lot of questions are still to be answered. Therefore, investigation of the interactions between rice and microbes for N metabolism is one of the promising future projects in terms of agronomic significance.

Adjusting phytohormone metabolism for N utilization, identifying key factors for rice-microbe interaction, and investigating the crosstalk between N and other nutritional molecules will be valuable strategies for future breeding. There is an urgent demand for future in-depth studies that develop integrated strategies combining the traditional forward and reverse genetics along with high-throughput genomics to explore novel key components involved in N metabolism and management at the whole plant level.

**Funding:** This research was funded by the Institute for Basic Science (IBS-R013-D1) from the Ministry of Science.

**Institutional Review Board Statement:** Not applicable.

**Informed Consent Statement:** Not applicable.

**Data Availability Statement:** Not applicable.

**Acknowledgments:** Thanks to Céline Masclaux-Daubresse (IJPB, INRAE Versailles, France) for proof reading the manuscript.

**Conflicts of Interest:** The author declares no conflict of interest.

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
