# Peer review of "Recent Advances on Nitrogen Use Efficiency in Rice"

_agronomy, doi:10.3390/agronomy11040753_

Round 1

Reviewer 1 Report

excellent paper . Could be acepted as it is 

Author Response

excellent paper. Could be accepted as it is 

Response: I am very thankful for your positive consideration. I revised several typos and reference lists throughout the text and have used the "Track Changes function in MS Word to make final edits.

Reviewer 2 Report

Dear Author,

The manuscript titled: “Recent advances on Nitrogen Use Efficiency in Rice” is an interesting review about efficient use of application of nitrogen (N) fertilization during growth of rice. This efficiency is essential for the development of sustainable agriculture and has become urgently needed. sustainable agriculture is very important topic. Therefore, I think that, this article could be interested for large scientific community. The presented manuscript is consistent and well written. The statements are clear. However, in my opinion thein not sufficient information about technical solutions which have been proposed to improve NUE in rice. The author limits himself only to making a statement. “Efforts to identify new targets for NUE improvement are remarkable in rice, and several technical solutions have been proposed to improve NUE in rice during the last years” Which in my opinion is too simplistic insufficient and this fragment requires further development. Authors also did not provide any information about arbuscular mycorrhizal, their great potential to reduce N loss from soil, and their role in efficient use of application of nitrogen (N) fertilization during growth of rice. I my opinion this information also should be added. Therefore in my opinion the presented manuscript could be published in Agronomy after minor revision.

Author Response

Reviewer#2

The manuscript titled: “Recent advances on Nitrogen Use Efficiency in Rice” is an interesting review about efficient use of application of nitrogen (N) fertilization during growth of rice. This efficiency is essential for the development of sustainable agriculture and has become urgently needed. Sustainable agriculture is very important topic. Therefore, I think that, this article could be interested for large scientific community. The presented manuscript is consistent and well written. The statements are clear. However, in my opinion thein not sufficient information about technical solutions which have been proposed to improve NUE in rice.

Response: I appreciate the reviewers’ comments. Throughout the paper, I have presented the recent strategies involved in the genetic manipulation of N uptake, assimilation and remobilization, and their regulation. In each section, I have selected the candidate genes and described the nitrogen-related phenotypes of their transgenic lines. All these results are summarized and listed in Table 1 and 2.

The author limits himself only to making a statement. “Efforts to identify new targets for NUE improvement are remarkable in rice, and several technical solutions have been proposed to improve NUE in rice during the last years” Which in my opinion is too simplistic insufficient and this fragment requires further development.

Response: Thanks for the critical comment. I revised the text as below.

Although a substantial identification of targets for NUE improvements and several technical strategies to improve NUE in rice have been conducted during the last years, problems are still emerging, including the complex trait of N metabolism, inconsistent outcomes from the transgenic approaches, and shortfall of test in the paddy field. To solve these problems properly, continuous efforts will be required to scientists by combining the molecular approaches and agricultural practices in an eco-friendly manner (page 11, line 36-42). I have used the "Track Changes function in MS Word to make final edits.

Authors also did not provide any information about arbuscular mycorrhizal, their great potential to reduce N loss from soil, and their role in efficient use of application of nitrogen (N) fertilization during growth of rice. I my opinion this information also should be added. Therefore, in my opinion the presented manuscript could be published in Agronomy after minor revision.

Response: Thanks for the valuable suggestion. As the reviewer indicated, information about arbuscular mycorrhizal has been important for sustainable agriculture. Although I just mentioned it briefly in the submitted paper at the ‘Future perspectives’ part, I revised the text as the reviewer requested as below (page 12, line 3-12).

- It is well known that root-associated microbes play key roles in the acquisition and cycling of nutrients including N [117]. N fertilization can affect the soil properties and activities of the rhizosphere microbial community, which have in turn significant influences rice production [118]. In addition, information on arbuscular mycorrhizal fungi (AMF) can be considered as a crucial tool for sustainable N fertilization, which play important role in the maintenance of soil fertility, the reduction of chemical fertilizers as well as crop production [119]. Research on this area is attracting attention as a rising research hotspot, but a lot of questions are still to be answered. Therefore, investigation of the interactions between rice and microbes for N metabolism is one of the promising future projects in terms of agronomic significance.

  1. Chen, J.; Arafat, Y.; Ud Din, I.; Yang, B.; Zhou, L.; Wang, J.; Letuma, P.; Wu, H.; Qin, X.; Wu, L.; et al. Nitrogen Fertilizer Amendment Alter the Bacterial Community Structure in the Rhizosphere of Rice (Oryza sativaL.) and Improve Crop Yield. Front Microbiol. 2019, 10, 2623.
  2. Giovannini, L.; Palla, M.; Agnolucci, M.; Avio, L.; Sbrana, C.; Turrini, A.; Giovannetti, M. Arbuscular Mycorrhizal Fungi and Associated Microbiota as Plant Biostimulants: Research Strategies for the Selection of the Best Performing Inocula. Agronomy202010, 106. 

Reviewer 3 Report

The manuscript submitted in Journal agronomy entitled “Recent advances on Nitrogen Use Efficiency in Rice” has been critically reviewed. Rice is stable cereal grain for more than 50% of world population. Rice is cultivated and consumed mainly in Asian countries and nitrogen management play significant role in its economical production. This manuscript cover nitrogen use efficiency (NUE) in rice crop and if feel such study has global significance. Therefore, I recommend this manuscript for publication. I have one minor suggestion in order update present manuscript.

Minor comment:  Kindly write briefly about different approaches to enhance NUE such as agronomic, breeding etc., in rice if possible for author.   

Author Response

Reviewer#3

The manuscript submitted in Journal agronomy entitled “Recent advances on Nitrogen Use Efficiency in Rice” has been critically reviewed. Rice is stable cereal grain for more than 50% of world population. Rice is cultivated and consumed mainly in Asian countries and nitrogen management play significant role in its economical production. This manuscript cover nitrogen use efficiency (NUE) in rice crop and if feel such study has global significance. Therefore, I recommend this manuscript for publication. I have one minor suggestion in order update present manuscript. Minor comment: Kindly write briefly about different approaches to enhance NUE such as agronomic, breeding etc., in rice if possible for author.

Response: I thank you for the valuable comment. I include the sentence below at the end of the second section (What is the Nitrogen use efficiency?). I have used the "Track Changes function in MS Word to make final edits.

“Strategies to improve NUE in rice have mainly involved in the genetic modification of N uptake, N assimilation and remobilization and their regulations [7]. In addition, conventional breeding, agronomic managements such as the method of applying N fertilizes and cropping system, and combination of them is also critical to increase NUE [12]” (page 2, line 28-32). I have used the "Track Changes function in MS Word to make final edits.
